# Current Surgical Concepts and Indications in the Management of the Short Bowel State: A Call for the Use of Multidisciplinary Intestinal Rehabilitation Programs

**DOI:** 10.3390/children8080654

**Published:** 2021-07-29

**Authors:** Antonino Morabito, Sara Ugolini, Maria Chiara Cianci, Riccardo Coletta

**Affiliations:** 1Department of Pediatric Surgery, Meyer Children’s Hospital, University of Florence, 50139 Florence, Italy; 2Department of Neuroscience, Psychology, Drug Research and Child Health (NEUROFARBA), University of Florence, 50121 Florence, Italy; s.ugolini@live.it (S.U.); mariachiara.cianci@unifi.it (M.C.C.); riccardo.coletta@meyer.it (R.C.); 3School of Heath and Society, University of Salford, Manchester M6 6PU, UK

**Keywords:** short bowel syndrome, child, infant, intestinal diseases, intestinal failure, autologous gastrointestinal reconstruction, parenteral nutrition

## Abstract

The mainstay of management for short bowel syndrome (SBS) is to promote access to the best quality of care provided by the intestinal rehabilitation program (IRP) in specialized centres. When treating SBS patients, the main goal is to minimize disease-associated complications, as well as achieve enteral autonomy. Surgical strategies should be selected cautiously upon the actual state of the bowel with respect to what it is clinically relevant for that specific patient. To this aim, a personalized and multidisciplinary approach for such a complex syndrome is needed.

## 1. Introduction

Short bowel syndrome (SBS) is a multi-system disorder caused by malabsorption of nutrients as a result of inadequate intestinal length [1,2]. The overall survival has increased over the last decades, even with significant losses of bowel length [3]. This has mainly been due to improved overall care, new total parenteral nutrition (TPN) formulas, and improved surgical techniques [4,5]. However, TPN complications are still a challenge for rehabilitation teams and long-term parenteral nutrition (PN) may lead to intestinal failure-associated liver disease (IFALD) [6,7]. The final goal of treatment in these patients should be to achieve enteral autonomy (EA), minimizing complications. Bowel lengthening procedures have increasingly been proposed for long-term TPN patients who fail to improve intestinal function with non-invasive strategies and, in selected groups of patients, bowel transplantation can be an option to consider [6,8,9]. The aim of non-transplant surgery is to restore normal (or close to normal) physiology by optimizing the absorptive surface area, improving peristalsis and decreasing transit time by reshaping morphology (i.e., narrowing and lengthening the bowel by taking advantage of pathologic dilatation). Clear indications and appropriate timing of the procedures have not been fully determined and understood. Starting from our own experience, according to the intestinal rehabilitation program (IRP), we aimed to review the available knowledge in order to delineate steps of management for autologous gastrointestinal reconstruction (AGIR) in general and lengthening procedures [10,11,12].

## 2. Intestinal Rehabilitation Program

Because of the multi-systemic nature and complexity of the disease, IRP is founded on a multidisciplinary team, including paediatric surgeons, gastroenterologists, nutritionists, logopedists and specialised nurses [2,3,13,14]. As is widely known, early entry into an IRP at one of the highly specialized centres is crucial for survival for SBS patients [12,13,15,16,17,18]. At the time of first assessment, data for personalized management are collected (neonatal history, surgical history, nutritional status, status of the anatomy and physiology of the remaining bowel, and potential metabolic and/or central line complications) [19,20,21,22]. In fact, a 360-degree multidisciplinary evaluation of the patient state is key to orienting management. Within an IRP, liver-sparing PN is commenced to lower the risk of the alarming IFALD [4,5,6]. Moreover, oral feeding is usually preferred since it more closely resembles the natural physiology, stimulating the bowel mucosa, enhancing adaptation and promoting brain learning. When possible, naso-gastric tubes or gastrostomy feeding are avoided [23]. Feeding is essential and, for the purposes of AGIR purposes, it encourages the natural process of bowel adaptation, which may ultimately lead to proficient surgical handling of dilated bowel [9,24].

## 3. Autologous Gastrointestinal Reconstruction (AGIR)

AGIR represents a systematic approach to SBS with the aim to optimize both patient and bowel and it should be rigorously planned within the context of an IRP program [1,25,26,27,28,29]. As already said, in IRP Centers, a teamwork-based process is promoted to achieve EA, which, in terms of anatomy optimization, may require more than one surgical procedure. One key concept in AGIR is the proficient utilization of “bowel dilatation”, which is a mechanism of physiological adaptation and, eventually, also a cause of added morbidity due to bacterial overgrowth and translocation. Distraction enterogenesis techniques such as controlled tissue expansion (CTE) stimulate the growth of all the layers, including the mucosa, thus providing material for further lengthening procedures [29,30,31]. The first line surgical treatment following an extensive small bowel resection is, in fact, “minimal surgery”, which is central line insertion for PN and stoma formation. This allows fecal recirculation for mucosal trophic purposes (from the proximal stoma to the distal one) and “artificial” bowel dilatation thanks to the intermittent partial obstruction of the proximal stoma [2,29,32]. When the patient is planned for the reconstruction, various previously described techniques are available: longitudinal intestinal lengthening and tailoring (LILT), spiral intestinal lengthening and tailoring (SILT), serial transverse enteroplasty (STEP), colonic interposition and reversed segments [1,2,3,10,11,12,29,32,33,34,35,36,37,38,39]. The choice should be made in a personalized manner by careful evaluation of the specific patient, anatomy and physiology. According to practice, the main indications for surgery in general are severe SBS, bowel dilatation, bacterial overgrowth and failure to progress to EA. Patients that would possibly benefit from AGIR can be identified among three groups: 1: severe SBS (neonates with 5–20 cm bowel length); 2: clinically stable “PN-children” presenting with distended loops of bowel; 3: “non-dilated SBS” patients with a rapid transit time. For severe SBS, the early introduction into an IRP is crucial to facilitate bowel adaptation with a consequent reduction in PN-dependence [2,3]. In these patients, a careful evaluation of the anatomy is mandatory as they may require multiple or combined procedures. Patients with >40 cm of bowel and an intact ileo-caecal valve (ICV) are expected to achieve EA by simple adaptation but could still benefit from some procedures to delay the transit time and to reduce the bacterial overgrowth [33,40]. Any degree of bowel dilatation should be considered for surgery if clinically significant [9]. Patients with “non-dilated SBS” with fast transit time are also considered eligible for lengthening with the goal of improving the remaining bowel physiology. This group of patients may benefit from anti-peristaltic reversed segments and iso- or anti-peristaltic colonic interposition [41].

## 4. Our Experience

After the Manchester experience [3], the authors started the IRP at the Meyer Children Hospital in Florence in mid-2018. Thirty consecutive children were treated for intestinal failure due to SBS. The program is based on a multidisciplinary team of paediatric surgeons, gastroenterologists, dieticians, neonatologists, specialised nurses and psychologists. Patients underwent primary assessment followed by a personalized plan of intestinal rehabilitation, which was discussed by panels of nutritionists, dieticians, speech and language therapist and surgeons. Patients with clinical features such as bowel dilatation or fast transit time, as previously described, underwent surgical reconstruction within the IRP. For these patients, the choice of the preferred technique (or the combination of them) was based on preoperative assessment of actual bowel anatomy and function and patients’ general clinical status. Among the patients we recorded 15 females (50%) and a median age at presentation of 3.5 years (range 0.1–18.6 years), Q1 = 1; Q3 = 9.5; IQR = 8.5. Diseases underlying SBS were neonatal volvulus (8/30 patients, 26.6%), necrotizing enterocolitis (7/30 patients, 23.3%), intestinal atresia type 4 (4/30 patients, 13.3%), gastroschisis (4/30, 13.3%), Hirschsprung disease (3/30, 10%), ulcerative colitis (2/30, 6.6%), and obstruction with perforation (2/30, 6.6%). All these patients were presented at our centre with parenteral support of 5 nights on TPN (range 0–7), Q1 = 5; Q3 = 7; IQR = 2

Sixteen (53.3%) patients underwent autologous gastrointestinal reconstructive procedure where SILT was the most frequently performed procedure (6/30, 20%). The median bowel length at the time of initial surgery was 37.5 cm (range 15–345 cm), Q1 = 28; Q3 = 42.5; IQR = 14.5. A STEP was required in 3/30 patients (10%), a colonic reversed segment in 4/30 (13.3%), a Ziegler myotomy in one patient with a total intestinal aganglionosis (3.3%), a combination of SILT and the Mikulitz stricturoplasty technique was deemed necessary in one patient (3.3%); interestingly, LILT was performed only in one patient (3.3%). Final intestinal length after AGIR was 45.5 cm (range 15–345 cm), Q1 = 39.75; Q3 = 62.75; IQR = 23.

Postoperatively, nutritional intake was administered by continuous TPN for 5–7 days followed by gradual oral feeding and PN-cyclization. Within the series, the median follow-up was 24.5 months (range 7–35 months), Q1 = 17.75; Q3 = 28.25; IQR = 10.5. Five out of thirty (16.6%) patients were followed-up by distance because they resided in other cities/countries and 2/30 were lost to follow-up. After AGIR procedure, 13/29 (44.8%) were weaned off TPN, while 16/29 (55.2%) are progressing along their rehabilitation program reducing parenteral nutrition progressively with a mean of 3 nights on TPN (range 2–5), Q1 = 2.5; Q3 = 5; IQR = 2.5. In our series two patients only were referred for assessment in a transplant centre, of which a 2-month-old baby with intestinal atresia type 4 and severe hepatic fibrosis and a 10-year-old baby affected from central venous catheter related thrombosis of more than two central veins.

At follow-up, recorded complications were acute renal failure (ARF) (*n* = 1), wound dehiscence (*n* = 2), entero-cutaneous fistula (*n* = 1) and intestinal subocclusion (*n* = 2). Among those requiring therapy, ARF and subocclusion were treated by pharmacological interventions while the entero-cutaneous fistula was initially treated by VAC-therapy and then fibrin glue instillation as a second approach.

## 5. Discussion

The goal of the management of SBS is to achieve the best quality of life for patients, optimizing EA and decreasing the on-PN time. Children with SBS suffer from intestinal failure and its associated complications, with a dramatic impact upon quality of life and healthcare. Nutritional deficiency, the risk of bacterial overgrowth, infections and IFALD are subsequent disorders these patients must deal with, and the primary target of an IRP Centre. Various multidisciplinary strategies may be required and, in some cases, multiple surgeries. The choice of the specific surgical technique among the spectrum available is patient-centered, where the surgeon assesses the actual function and anatomy of the bowel (i.e., contrast study) in respect to what is clinically relevant for that specific patient. Which operation is required is based on the length of remaining bowel and the likelihood of the patient achieving EA. It is usually appropriate to choose a treatment algorithm that permits further surgeries thereafter (i.e., sequential lengthening), in order to keep intestinal function as physiological as possible [34]. The surgeons’ technical preferences should not interfere with the most suitable decision. In our practice, the LILT procedure is excellent as it doubles (100%) the length of the remaining bowel while respecting the anatomical criteria of bowel vascularization. This technique creates an optimal intestinal lumen and does not prevent any future bowel reconstructive surgeries if needed [35]. In fact, single or multiple anti-peristaltic (reversed) segments and colonic interposition can be added to further delay transit, increase mucosal contact time and enhance bowel adaptation and absorption [36]. Additional bowel length (68%) can be achieved through the STEP procedure [37]. The Iowa segment as described by Kimura could be of help as well [38]. It is important to consider the use of the remaining colon if present, performing a longitudinal colonic lengthening with or without a sigmoid J-pouch as described by Devesa et al. [39]. It is therefore crucial that SBS patients are referred to an intestinal rehabilitation program in specialized centers where surgical, medical, nutritional and psychological aspects are supported by a multidisciplinary team. Parents and patients should be offered all the available treatment possibilities in order to make a real informed decision. Only when they fully understand the options and the possible complications to each approach can they be enrolled in a specific program. If parents choose an IRP, patients are always made aware that one operation may not be curative. Despite the appropriate approach to SBS, only 70% of infants achieve EA [42]. A parallel could be drawn with the Kasai procedure [41], which is considered for biliary atresia in the first instance and a successful approach in 30–40% of cases, while liver transplant is advocated as a rescue procedure. In the case of SBS patients, we propose a similar surgical attitude, suggesting a treatment algorithm based on a combined medical and surgical approach. Transplant should be used only if patients fail to achieve intestinal adaptation once AGIR options have been exhausted, develop irreversible liver failure or are about to lose all central venous access [29,40,43].

## 6. Conclusions

The mainstay of management for a complex syndrome such as SBS is to promote access to the best quality of care, namely IRP Centres, where a disease-specific and personalized “patient-centred” approach improves survival and costs (e.g., reduced hospitalization and time on PN). Moreover, within an IRP, defined treatment algorithms with clear and shared indications and timing for surgery greatly impact outcomes such as survival rate and quality of life and a better selection of patients for transplantation ultimately improve organ utilization.

## Data Availability

Not applicable.

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
