# Peer review of "Current Surgical Concepts and Indications in the Management of the Short Bowel State: A Call for the Use of Multidisciplinary Intestinal Rehabilitation Programs"

_children, 2021, doi:10.3390/children8080654_

Round 1
Reviewer 1 Report
This essay discusses and highlights the importance of a personalized and multi-disciplinary approach for the management of children with short bowel syndrome (SBS). It focuses on the current autologous gastrointestinal reconstruction (AGIR) and bowel lengthening procedures available as surgical management for these patients.
The manuscript is well written and covers a relevant topic as the overall survival of children with more complex forms of short bowel syndrome has improved in the past years, and with it the long-term complications of the condition.
The authors review their own experience since creation of their Intestinal Rehabilitation Program (IRP) in 2018, and their results shows the benefits of the multi-disciplinary approach, with half the patients followed up coming off or reducing their parenteral nutrition (PN) support requirements. However it was unclear from their data if these outcomes were a result of the multidisciplinary approach or surgical interventions.
The authors’ experience data should preferably be presented as medians with IQRs, instead of means. Bowel length measurement ‘at presentation’ should be better defined, do the authors mean at the time of initial surgery? Would be interesting to know what the indications for AGIR were, and if there were any complications from surgery during the follow-up period. Data showing the PN requirements before and after surgical intervention would be informative to assess the effect of surgery.
Other minor comments/suggestions:
Introduction:
- Authors should be cautious in stating that ‘bowel lengthening procedures have increasingly been proposed to long-term TPN’, especially in the coming era of effective medical treatments such as GLP2 analogues, as not every patient would benefit from surgery.
Intestinal Rehabilitation Program:
- SBS-associated liver disease should be replaced by intestinal failure-associated liver disease (IFALD).
Discussion:
- Authors state that ‘in our practice, the LILT procedure is excellent as it doubles the length of the remaining bowel’. However in their experience results they report that SILT was the most frequently performed procedure (7/16 patients undergoing bowel lengthening procedures). What were the reasons from favouring SILT over LILT?
References:
- Reference number 24 is duplicated
- Several references are incomplete (ref 6, 11, 12, 16, 37, 42)
Author Response
Thank you very much for your kind and useful suggestions. I will answer point by point as requested.
Point 1: The authors review their own experience since creation of their Intestinal Rehabilitation Program (IRP) in 2018, and their results shows the benefits of the multi-disciplinary approach, with half the patients followed up coming off or reducing their parenteral nutrition (PN) support requirements. However it was unclear from their data if these outcomes were a result of the multidisciplinary approach or surgical interventions.
Response 1: Thank you very much for pointing out this issue. Our paper (and our vision) wants to put into evidence the concept of the multidisciplinary of an IRP and the benefit for the patient of this management. Together with this, we wanted to briefly report about the patients that needed a surgical intervention. We indeed have changed the text and we really hope that the new version better reflects our aims: "Patients underwent primary assessment followed by a personalized plan of intestinal rehabilitation which was discussed at panels of nutritionists, dieticians, logopedists and surgeons. Patients complicated by clinical features such as bowel dilatation or fast transit time, as previously described, underwent surgical reconstruction within the IRP; For them, with the choice of preferred technique (or the combination of them) was based on preoperative assessment of a combination of different surgical strategies depending on the actual bowel anatomy and function , and patients’ general clinical status."
Point 2: The authors’ experience data should preferably be presented as medians with IQRs, instead of means. Bowel length measurement ‘at presentation’ should be better defined, do the authors mean at the time of initial surgery? Would be interesting to know what the indications for AGIR were, and if there were any complications from surgery during the follow-up period. Data showing the PN requirements before and after surgical intervention would be informative to assess the effect of surgery.
Response 2: Thank you very much. We provided data as suggested. For patients who reside outside the dominion of our regional health care system (such as other countries, i.e Brasil or Greece), gastroenterologists, nutritionists and dietitians regularly receive informations from the local caregivers about the trend of PN requirements. Unfortunately, at the center, at the time of these periodical evaluations, the nutrients and fluids amounts and the calories administered by PN haven't been recorded. The data of the follow-up refer to the number of patients off-PN, those who have gradually been decreasing the PN-amount per week and those who are not improving. The aim of our paper is not to review our case series but to suggest the validity of a multidisciplinary management in a specialized centre for the treatment of SBS patients.
Point 3: Authors should be cautious in stating that ‘bowel lengthening procedures have increasingly been proposed to long-term TPN’, especially in the coming era of effective medical treatments such as GLP2 analogues, as not every patient would benefit from surgery.
Response 3: Thank you, we added this important clarification to the text.
Point 4: - SBS-associated liver disease should be replaced by intestinal failure-associated liver disease (IFALD).
Response 4: Thank you, we proceeded to that replacement.
Point 5: Authors state that ‘in our practice, the LILT procedure is excellent as it doubles the length of the remaining bowel’. However in their experience results they report that SILT was the most frequently performed procedure (7/16 patients undergoing bowel lengthening procedures). What were the reasons from favouring SILT over LILT?
Response 5: Thank you for pointing this out to us. SILT and LILT have different indications. (Ref: Restoring gut physiology in short bowel patients: from bench to clinical application of autologous intestinal reconstructive procedures. Expert Rev Gastroenterol Hepatol. 2019 Aug.) The choice of the procedure is based on the quality of the mesenteric and the degree of bowel’s dilatations. This is the reason why one procedure had been chosen over the other.
Point 6: References: - Reference number 24 is duplicated - Several references are incomplete (ref 6, 11, 12, 16, 37, 42).
Response 6: I am very sorry for this. We eliminated the reference 24, which was in fact already cited as number 9. For the "incomplete" ones, we double checked the citation methods, but we couldn't identify any error.
We hope this is sufficient to clear your queries Thank you again for the review
Reviewer 2 Report
This is a perspective on a highly important topic.
To have the audience benefits the most from this important work, I believe the manuscript will need major modifications.
Authors currently provide general perspective on the IRP and general views on their patient management. Although this can be useful information, to help the audience gather practical information, there is need for structured information with take home messages. One possible strategies could be for authors to provide a case series of their center following the standard of reporting and then put that into perspective of having a standardized approach and define the suggested standardized approach. If authors were to recommend a guideline/ strategy pathway for patients with this disease, what would that be?
There is no definition of population. I assume pediatric but hat has to be specified.
I do believe the manuscript benefits from English editing.
Author Response
Thank you very much for your kind and useful suggestions. I will answer point by point as requested.
Point 1: The Authors currently provide general perspective on the IRP and general views on their patient management. Although this can be useful information, to help the audience gather practical information, there is need for structured information with take home messages. One possible strategies could be for authors to provide a case series of their center following the standard of reporting and then put that into perspective of having a standardized approach and define the suggested standardized approach. If authors were to recommend a guideline/ strategy pathway for patients with this disease, what would that be?
Response 1: Thank you very much for pointing out this issue. Our paper wants to highlight MDT approsch to IRP and reasons why to refer to it. Our previously and published papers have focused on the guidelines and the importance of the intestinal rehabilitation program and have provided a practical pathway: Khalil BA, Ba'ath ME, Aziz A, Forsythe L, Gozzini S, Murphy F, Carlson G, Bianchi A, Morabito A. Intestinal rehabilitation and bowel reconstructive surgery: improved outcomes in children with short bowel syndrome. J Pediatr Gastroenterol Nutr. 2012 Apr;54(4):505-9. doi: 10.1097/MPG.0b013e318230c27e. PMID: 21832945.; Ba'ath ME, Almond S, King B, Bianchi A, Khalil BA, Morabito A. Short bowel syndrome: a practical pathway leading to successful enteral autonomy. World J Surg. 2012 May;36(5):1044-8. doi: 10.1007/s00268-012-1512-5. PMID: 22374542.
Point 2: There is no definition of population. I assume pediatric but hat has to be specified.
Response 2: Thank you very much. When speaking about concepts of IRP and AGIR, these can be applied to the adult SBS also. Concerning "our experience" section, it is clearly stated at the beginning that we operate at a children's hospital and that 30 consecutive children have been treated there.
Point 3: I do believe the manuscript benefits from English editing.
Response 3: Thank you, we checked the language with a service of editing.
We hope this is sufficient to clear your queries Thank you again for the review

Round 2
Reviewer 2 Report
I read with interest the following submitted work to Children-MDPI. In my first round of review, I had a few thoughts on how this work could be enhanced. I can see in the reply letter that my suggestions , unfortunately, have not been of much use to the authors. I do acknowledge the first author of this work has experience and previous important publications in the field of pediatric SBS. I, however, believe the current work that is submitted to the Children MDPI journal lacks clear message and structure.
Author Response
Thank you very much for your careful reading of our text. We very much appreciate your suggestions. We have made a considerable effort to take into account the interesting suggestions proposed. We structured the manuscript as a "perspective type", meaning to offer a clear, simple and brief idea of the reasons why to refer to an IRP center.
sincerely